# PDE2A2 regulates mitochondria morphology and apoptotic cell death via local modulation of cAMP/PKA signalling

Stefania Monterisi[1,2], Miguel J Lobo[1,2], Craig Livie[3], John C Castle[1,2], Michael Weinberger[1,2], George Baillie[4], Nicoletta C Surdo[1,2], Nshunge Musheshe[5], Alessandra Stangherlin[3], Eyal Gottlieb[6], Rory Maizels[1,2], Mario Bortolozzi[7,8], Massimo Micaroni[9], Manuela Zaccolo[1,2]*

[1]Department of Physiology Anatomy and Genetics, University of Oxford, Oxford, United Kingdom; [2]BHF Centre of Research Excellence, University of Oxford, Oxford, United Kingdom; [3]Institute of Neuroscioence and Psychology, University of Glasgow, Glasgow, United Kingdom; [4]Institute of Cardiovascular and Medical Science, University of Glasgow, Glasgow, United Kingdom; [5]Department of Molecular Pharmacology, University of Groningen, Groningen, The Netherlands; [6]Beatson Institute, University of Glasgow, Glasgow, United Kingdom; [7]Department of Physics and Astronomy "G. Galilei", University of Padova, Padova, Italy; [8]Venetian Institute of Molecular Medicine, University of Padova, Padova, Italy; [9]Swedish National Centre for Cellular Imaging, Sahlgrenska Academy, University of Gothenburg, Gothenburg, Sweden

*For correspondence: manuela.
zaccolo@dpag.ox.ac.uk

Competing interests: The
authors declare that no
competing interests exist.

Reviewing editor: Tony Hunter,
Salk Institute for Biological
Studies, United States

**Abstract** cAMP/PKA signalling is compartmentalised with tight spatial and temporal control of signal propagation underpinning specificity of response. The cAMP-degrading enzymes, phosphodiesterases (PDEs), localise to specific subcellular domains within which they control local cAMP levels and are key regulators of signal compartmentalisation. Several components of the cAMP/PKA cascade are located to different mitochondrial sub-compartments, suggesting the presence of multiple cAMP/PKA signalling domains within the organelle. The function and regulation of these domains remain largely unknown. Here, we describe a novel cAMP/PKA signalling domain localised at mitochondrial membranes and regulated by PDE2A2. Using pharmacological and genetic approaches combined with real-time FRET imaging and high resolution microscopy, we demonstrate that in rat cardiac myocytes and other cell types mitochondrial PDE2A2 regulates local cAMP levels and PKA-dependent phosphorylation of Drp1. We further demonstrate that inhibition of PDE2A, by enhancing the hormone-dependent cAMP response locally, affects mitochondria dynamics and protects from apoptotic cell death.

## Introduction

Mitochondria produce the majority of ATP required for a cell to function and perform vital functions in the maintenance of cellular metabolism and ion homeostasis. Additionally, mitochondria play an important role in apoptosis and are implicated in the pathogenesis of many diseases (*Nunnari and Suomalainen, 2012*). Mitochondria exist in a dynamic network and are continuously remodelled by fusion and fission reactions. Alteration of the fusion/fission balance contributes to the pathogenesis of many complex conditions, including common neurodegenerative diseases, cancers and cardiovascular disorders (*Archer, 2013*), including the adaptive response to ischaemia–reperfusion injury

(*Ong et al., 2010*; *Sharp et al., 2014*) and cardiac remodelling associated with heart failure (*Chen et al., 2009*). Disorganized, small mitochondria are typically found in a variety of cardiac pathologies (*Schaper et al., 1991*; *Chen et al., 2009*). As a consequence, molecular mediators of mitochondria dynamics are recognised as potential therapeutic targets (*Archer, 2013*).

Mitochondria fission involves dynamin-related protein 1 (Drp1), a GTPase of the dynamin superfamily, which resides in the cytosol and translocates to the mitochondria upon activation by calcineurin-dependent dephosphorylation (*Cereghetti et al., 2008*). Drp1 multimerises at the outer mitochondrial membrane (OMM) and is thought to mechanically constrict and eventually sever mitochondria. Drp1 is subject to complex post-translational modification by ubiquitylation, sumoylation, nitrosylation and phosphorylation. A well-characterised regulation of Drp1 is its inactivation by protein kinase A (PKA)-dependent phosphorylation at serine 637 (ser637), which results in mitochondria elongation (*Cereghetti et al., 2008*; *Cribbs and Strack, 2007*; *Chang and Blackstone, 2007*).

PKA is a multi-target kinase activated by the ubiquitous second messenger 3',5'-cyclic adenosine monophosphate (cAMP). cAMP is synthesised either by a plasma membrane- associated adenylyl cyclase (pmAC), upon hormonal activation of Gs protein-coupled receptors, or by a $Ca^{2+}$ and bicarbonate sensitive soluble adenylyl cyclase (sAC) (*Rahman et al., 2013*). cAMP/PKA signalling regulates fundamental cellular processes, including cell differentiation, growth, metabolism and death (*Taskén and Aandahl, 2004*). Dysfunctional cAMP signalling has been implicated in multiple disease conditions and several drugs currently in use target the cAMP/PKA pathway. cAMP/PKA signalling is compartmentalised in distinct signalling domains and occurs largely via generation of restricted pools of cAMP that activate PKA subsets tethered in proximity to specific targets via binding to A kinase anchoring proteins (AKAPs) (*Langeberg and Scott, 2015*).

Phosphodiesterases (PDEs) constitute a superfamily of enzymes, which includes more than 100 isoforms, and are the only enzymes that degrade cAMP. Different PDE isoforms are uniquely regulated and distributed within the cell. Therefore, they differentially determine the local level of cAMP at specific subcellular sites, dictating which PKA targets are phosphorylated and the specificity of the downstream response (*Maurice et al., 2014*).

A number of components of the cAMP signalling cascade have been located at the mitochondria, including multiple AKAPs (*Huang et al., 1999*; *Alto et al., 2002*; *Means et al., 2011*) and PDEs (*Cercek and Houslay, 1982*; *Shimizu-Albergine et al., 2012*; *Acin-Perez et al., 2011*), suggesting the co-existence at the organelle of multiple cAMP/PKA signalling domains (*Lefkimmiatis and Zaccolo, 2014*). However, the organisation, regulation and functional significance of these domains remain largely to be established.

PDE2A is a 3',5'-cyclic guanosine monophosphate (cGMP)-activated PDE that degrades both cAMP and cGMP (*Stroop and Beavo, 1991*) and is expressed in a number of tissues, including brain, heart, liver, lung, adipose tissue and adrenal gland. Three variants of the Pde2a gene are expressed (PDE2A1, PDE2A2 and PDE2A3). The variants differ in their amino termini, and this variation may explain their different subcellular localisations (*Lugnier, 2006*). Genetic ablation of PDE2A results in high embryonic lethality past E17.5–E18.5 dpc (*Stephenson et al., 2009*), indicating that these enzymes are involved in vital biological functions.

Previous evidence suggests localisation of PDE2A2 to the mitochondrial matrix where it regulates ATP production via modulation of cAMP generated locally by sAC (*Acin-Perez et al., 2011*). Here, we demonstrate that in cardiac myocytes and other cell types PDE2A2 is part of a distinct cAMP/PKA signalling domain located at the mitochondria but outside the mitochondrial matrix. This PDE2A2 subset localises to mitochondrial membranes where it controls a pool of cAMP generated at the plasma membrane by the hormone-responsive pmAC, and regulates PKA-dependent phosphorylation of Drp1. Inhibition of this subset of PDE2A2 results in elongated mitochondria and protection from mitochondrial-dependent cell death.

## Results

### PDE2A2 regulates mitochondria morphology

When we expressed PDE2A2-GFP in primary neonatal rat ventricular myocytes (NRVM), we found that it is distinctly located at the mitochondria, unlike PDE2A1-GFP, which is cytosolic, and PDE2A3-GFP, which localises predominantly to the plasmalemma (*Figure 1A*). We also observed that

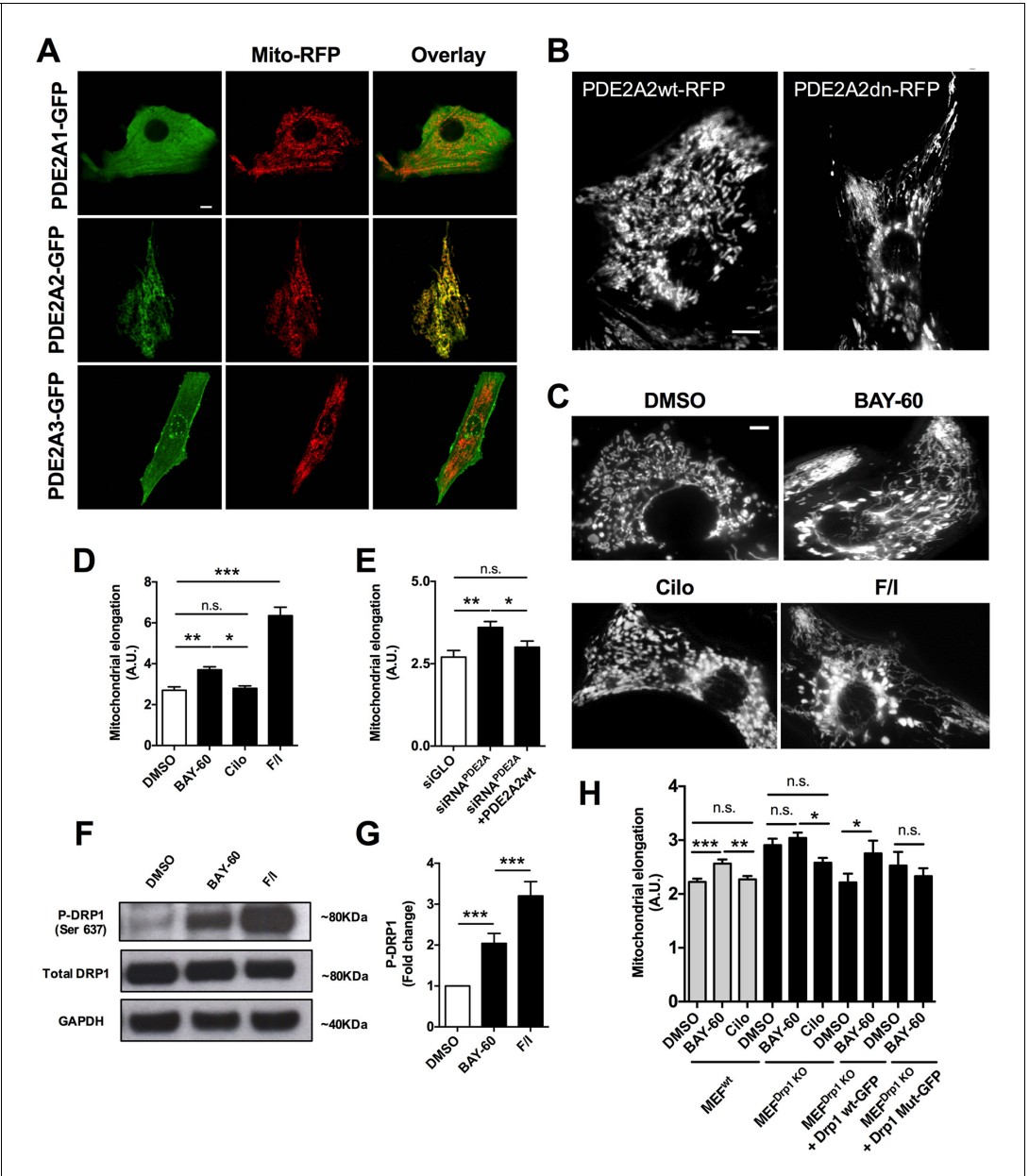

**Figure 1.** PDE2A2 localises to the mitochondria in NRVM and regulates mitochondria morphology via DRP1. (**A**) Localisation of PDE2A1-GFP, PDE2A2-GFP and PDE2A3-GFP in neonatal rat ventricular myocytes (NRVM) labelled with mitotracker red. Scale bar: 10 µm. (**B**) Localisation of wild type (PDE2A2wt-RFP) or catalytically inactive (PDE2A2dn-RFP) PDE2A2 in NRVM. Scale bar: 10 µm. (**C**) NRVM treated with the indicated drugs (F/I: Forskolin/IBMX) and incubated with mitotracker red to stain mitochondria. Scale bar: 10 µm. (**D**) Quantitative analysis of mitochondrial length on cells as shown in C. n = 40 cells from three biological replicates. (**E**) Quantitative analysis of mitochondrial length in cells transfected with a control siRNA sequence (siGLO), a specific siRNA for PDE2A (siRNA$^{PDE2A}$) alone or in combination with a plasmid carrying a siRNA-resistant PDE2A2 sequence. n = 30 cells from 3 biological replicates. (**F**) Western blotting analysis of cell lysates obtained from NRVM treated with the indicated drugs and probed for phospho-DRP1 (ser637), total DRP1 and GAPDH, as indicated. Representative of 5 biological replicates. (**G**) Quantification of the western blotting analysis as shown in F. (**H**) Quantitative analysis of mitochondrial length in MEFs$^{wt}$ and MEFs $^{Drp1KO}$ stable clones treated with DMSO, BAY60 and Cilostamide, and MEFs$^{Drp1KO}$ overexpressing Drp1 wt-GFP or Drp1 mut -GFP and treated with the same drugs. n = 30 cells from three biological replicates. ANOVA test with Bonferroni correction was used for statistical analysis. *0.01≤p≤0.05, **0.001≤p<0.01, ***p<0.001.

The following figure supplements are available for figure 1:

**Figure supplement 1.** PDE2A2 regulates mitochondria morphology in H9C2 myoblasts.

**Figure supplement 2.** The effect of PDE2A inhibition on DRP1 phosphorylation at ser616.

expression of PDE2A2wt-RFP results in more fragmented mitochondria compared to expression of its catalytically inactive variant PDE2A2dn-RFP (*Figure 1B*).

To further investigate whether PDE2A2 has a role in the modulation of mitochondria morphology, we treated NRVM with the selective PDE2A inhibitor Bay 60–7550 (1 µM) and found significant mitochondrial elongation compared to DMSO-treated controls (*Figure 1C,D*). Increasing cAMP levels via selective inhibition of PDE3 with cilostamide (10 µM) had no effect (*Figure 1C,D*), whereas maximal elevation of cAMP, as achieved by application of the tmAC activator forskolin (25 µM) and the non-selective PDE inhibitor isobutyl-methyl-xanthine (IBMX, 100 µM), also resulted in mitochondria elongation (*Figure 1C,D*). Knock down of PDE2A expression in NRVM, using the small interfering RNA sequence siPDE2 (*Stangherlin et al., 2011*), also resulted in elongated mitochondria (*Figure 1E*). The specificity of the siRNA effect was confirmed by rescue of the mitochondrial phenotype upon overexpression of a siRNA-resistant PDE2A2wt-GFP (*Zoccarato et al., 2015*) (*Figure 1E*). In line with the above findings, treatment of the myoblast cell line H9C2 with Bay 60–7550 resulted in more elongated mitochondria (*Figure 1—figure supplement 1A*), whereas expression of PDE2A2wt-GFP, but not expression of PDE4D7wt-GFP, resulted in mitochondria fragmentation (*Figure 1—figure supplement 1B*). Western blotting analysis of lysates from NRVM treated with Bay 60–7550 showed significant increase in Drp1 phosphorylation at the PKA site ser637 (*Figure 1F,G*) compared to DMSO-treated controls, whereas no increased phosphorylation was detected at ser616, a site phosphorylated by the kinase CDK1 (*Taguchi et al., 2007*) (*Figure 1—figure supplement 2*). No effect of Bay 60–7550 treatment on mitochondrial morphology was found in mouse embryonic fibroblasts (MEFs) in which DRP1 was genetically ablated (*Ishihara et al., 2009*) (MEFs$^{DRP1-/-}$) (*Figure 1H*). Expression of wild type Drp1 (Drp1 wtGFP) in MEFs$^{DRP1-/-}$, but not of a DRP1 mutant that cannot be phosphorylated by PKA (Drp1 Mut-GFP), rescued the effect of Bay 60–7550 treatment (*Figure 1H*), confirming that elongation of mitochondria on inhibition of PDE2A is mediated by PKA phosphorylation of Drp1.

The role of PDE2A2 in the regulation of mitochondrial morphology is further supported by analysis of MEFs obtained from E.13 dpc embryos that are homozygous for the genetic ablation of the PDE2A gene (*Stephenson et al., 2009*) (MEFs$^{PDE2KO}$). *Figure 2A,C* shows that MEFs$^{PDE2KO}$ have significantly more elongated mitochondria than the wild type counterpart (MEFs$^{wt}$). The MEFs$^{PDE2KO}$ mitochondrial phenotype could be completely rescued by expression of a red fluorescent protein (RFP)-tagged PDE2A2 (PDE2A2wt-RFP), whereas expression of a catalytically inactive mutant of PDE2A2 (PDE2A2dn-RFP) in MEFs$^{WT}$ resulted in elongated mitochondria (*Figure 2B,C*). It should be noted that both PDE2A2wt-RFP and PDE2A2dn-RFP localise to mitochondria with no appreciable cytosolic localisation (*Figure 2B*). This suggests that the rescue effect of PDE2A2wt-RFP in MEFs$^{PDE2KO}$ is the consequence of local cAMP hydrolysis at the organelle, and that elongation of mitochondria in MEFs$^{WT}$ expressing PDE2A2dn-RFP relies on a local increase in cAMP following displacement of the endogenous active mitochondrial PDE2A2 (*Stangherlin et al., 2011*; *Zoccarato et al., 2015*). MEFs$^{PDE2KO}$ cells treated with the PKA inhibitor H89 show complete reversal of the mitochondrial phenotype, confirming involvement of PKA downstream of PDE2A inhibition (*Figure 2D*).

## Mitochondrial PDE2A regulates cAMP at the outer mitochondrial membrane

It has been previously reported that PDE2A2 is found in the mitochondrial matrix, where it hydrolyses cAMP generated locally by sAC (*Acin-Perez et al., 2011*). However, the molecular machinery that regulates mitochondria fission/fusion, including Drp1, is not localised to the matrix and any cAMP generated in the matrix cannot escape this compartment as the inner mitochondrial membrane (IMM) is impermeable to it (*Di Benedetto et al., 2008*; *Acin-Perez et al., 2009*). It is therefore unlikely that the elongation of mitochondria we observe on inhibition of PDE2A depends on modulation of cAMP generated in the matrix. We therefore hypothesised that mitochondrial PDE2A2 may be located outside the matrix in a compartment where it can access and degrade cAMP generated by pmAC. To test this hypothesis, we measured cAMP levels using FRET-based reporters that are targeted to the OMM (OMM-H90) (*Lefkimmiatis et al., 2013*), to the mitochondrial matrix (matrix-H90) (*Lefkimmiatis et al., 2013*), or are free in the cytosol (H90) (*van der Krogt et al., 2008*). When expressed in NRVM the three sensors show the expected localisation (*Figure 3* and *Figure 3—figure supplement 1*). As shown in *Figure 3A*, inhibition of PDE2A generates a similar elevation of

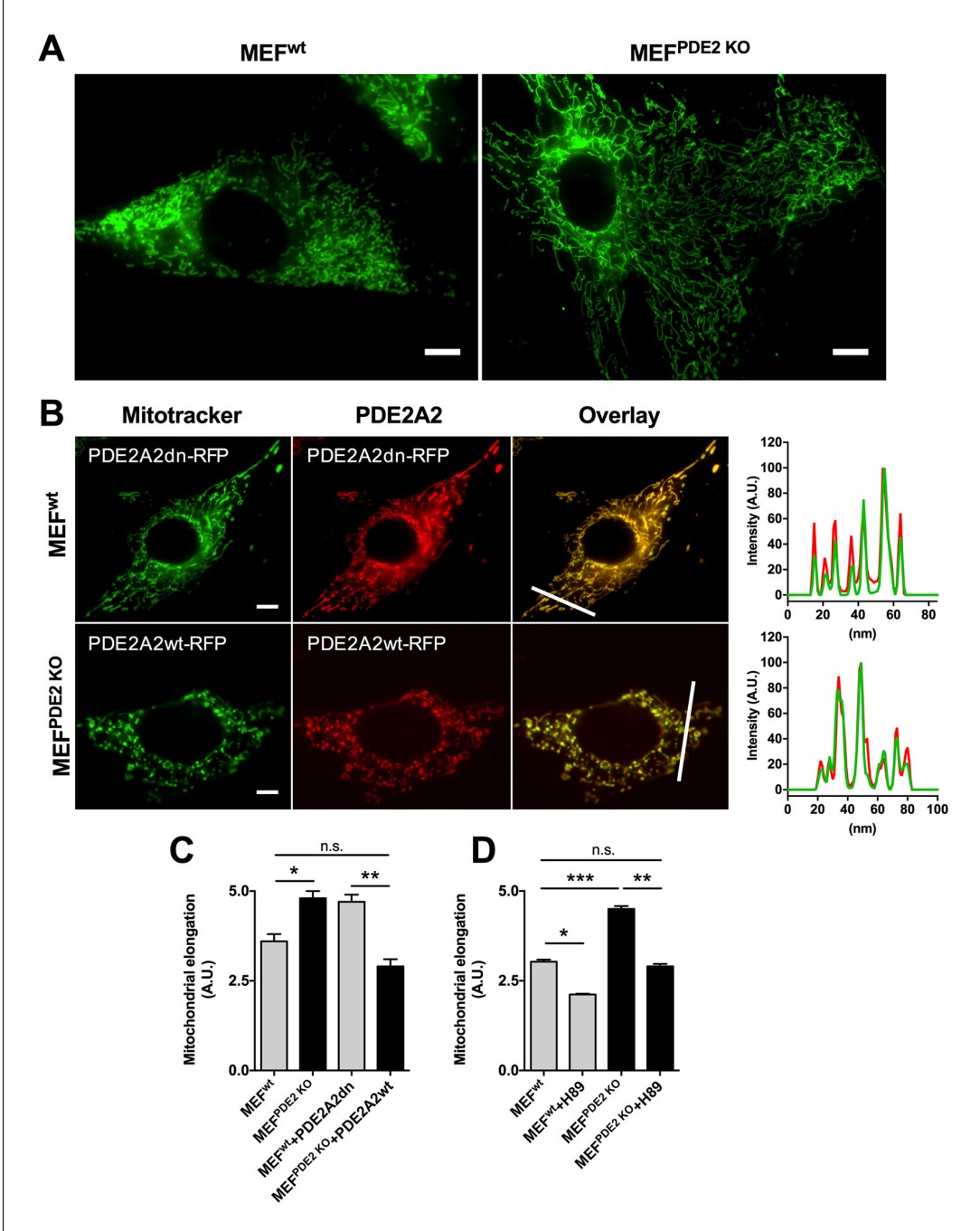

**Figure 2.** Mitochondria morphology is affected by PDE2A knock out. (**A**) Wild type (MEF^wt) and PDE2KO (MEF^PDE2KO) mouse embryonic fibroblasts stained with mitotracker green. Scale bar: 10 μm. (**B**) MEF^wt and MEF^PDE2KO expressing catalytically inactive (PDE2A2dn-RFP) or wild type (PDE2A2wt-RFP) PDE2A2, respectively and stained with mitotracker green. The overlay of the RFP and mitotracker signal is also shown. Panels on the right show the fluorescence intensity profile for the mitotracker (red line) and PDE2A2-RFP proteins (green line) along with the line shown in the overlay images. Scale bar: 10 μm. (**C**) Quantitative analysis of mitochondria morphology on images shown in **B**. n = 35 cells from three biological replicates. (**D**) Quantitative analysis of mitochondria morphology in MEF cells treated with the PKA inhibitor H89. n = 25 cells from two biological replicates. ANOVA test with Bonferroni correction was used for statistical analysis. *0.01≤p≤0.05, **0.001≤p<0.01, ***p<0.001.

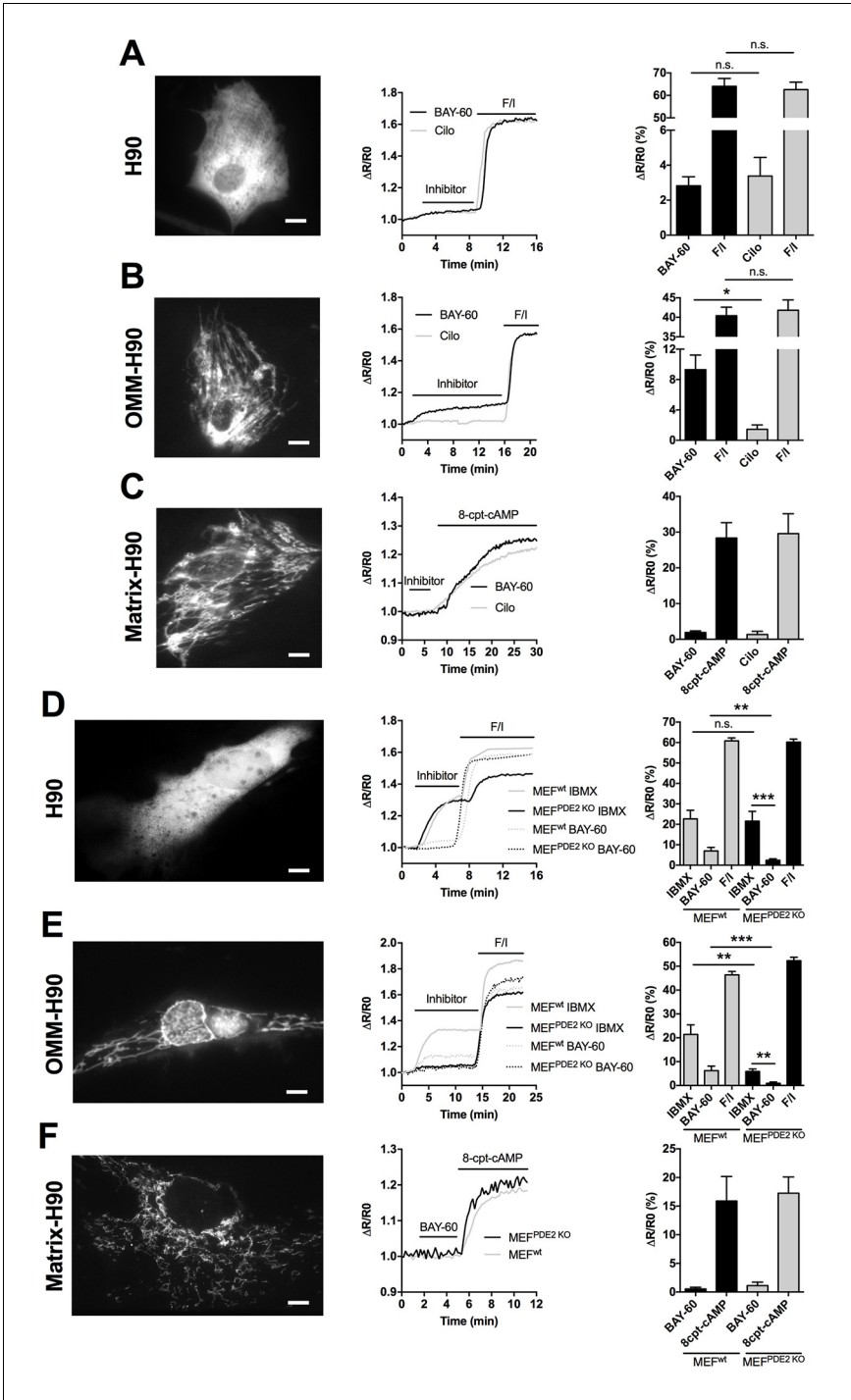

**Figure 3.** PDE2A2 controls cAMP at the outer mitochondrial membrane. (**A**) FRET analysis in NRVM expressing the cytosolic EPAC-based H90 sensor (left) and treated with the PDE2 selective inhibitor Bay 60–7550 (BAY60, 1 μM) or the PDE3 selective inhibitor cilostamide (Cilo), 10 μM). 25 μM Forskolin and 100 μM IBMX (F/I) was applied at the end of the experiment to achieve saturation of the sensor. Representative kinetics of FRET change are shown in the middle panel and summary of all experiments is shown in the panel on the right. n = 15 cells from three biological replicates. Scale bar: 10 μm. (**B**) FRET analysis in NRVM expressing an outer mitochondrial membrane-targeted version (OMM) of H90 and relative representative traces and summary of all the experiments. Cell treatment was as in **A**). n = 15 cells from three biological replicates. Scale bar: 10 μm. (**C**) FRET analysis in NRVM expressing a version of H90 targeted to the matrix and challenged with PDE selective inhibitors as in **A**, followed by application of the membrane permeable cAMP analogue 8CPT (200 μM). Representative kinetics and summary

*Figure 3 continued on next page*

*Figure 3 continued*

of experiments is shown in the middle and right panel, respectively. n = 12 cells from three biological replicates. Scale bar: 10 µm. (D) FRET analysis in Wild type (MEF$^{wt}$) and PDE2KO (MEF$^{PDE2KO}$) MEFs expressing the cytosolic EPAC-based H90 sensor (left) and treated with the non-selective PDEs inhibitor IBMX (100 µM) or the PDE2 selective inhibitor Bay 60–7550 (BAY60 100 nM). 25 µM Forskolin and 100 µM IBMX (F/I) was applied at the end of the experiment to achieve saturation of the sensor. Representative kinetics of FRET change are shown in the middle panel and summary of all experiments is shown in the panel on the right. n = 10 cells from three biological replicates. Scale bar: 10 µm. (E) FRET analysis in Wild type (MEF$^{wt}$) and PDE2KO (MEF$^{PDE2KO}$) expressing the outer mitochondrial membrane-targeted version (OMM) of H90 (left) and treated with the non-selective PDEs inhibitor IBMX (100 µM) or the PDE2 selective inhibitor Bay 60–7550 (BAY60 100 nM). Representative kinetics and summary of experiments is shown in the middle and right panel, respectively. n = 10 cells from three biological replicates. Scale bar: 10 µm. (F) FRET analysis in Wild type (MEF$^{wt}$) and PDE2KO (MEF$^{PDE2KO}$) expressing the mitochondrial matrix-targeted version of H90 (left) and treated with the PDE2 selective inhibitor Bay 60–7550 (BAY60 100 nM). Representative kinetics and summary of experiments is shown in the middle and right panel, respectively. n = 6 cells from two biological replicates. Scale bar: 10 µm. ANOVA test with Bonferroni correction was used for statistical analysis. *0.01≤p≤0.05, **0.001≤p<0.01, ***p<0.001.

The following figure supplements are available for figure 3:

**Figure supplement 1.** Localisation of cytosolic and targeted H90.

**Figure supplement 2.** Effect of PDE2A inhibition on cAMP levels in the matrix.

cAMP in the bulk cytosol as inhibition of PDE3. In contrast, and in line with the effect on mitochondria morphology (*Figure 1C,D*), a significantly higher increase in cAMP was detected at the OMM upon inhibition of PDE2A, as compared to inhibition of PDE3 (*Figure 3B*). Surprisingly, Bay 60–7550 did not increase cAMP levels in the matrix (*Figure 3C*) and was unable to appreciably increase in the matrix the concentration of the membrane permeable cAMP analogue 8-CPT-6-Phe-cAMP (*Figure 3—figure supplement 2*). When matrix-H90 was expressed in MEFs$^{PDE2wt}$ no appreciable increase in cAMP was detected in the matrix on inhibition of PDE2A and the response was undistinguishable from that obtained in MEFs$^{PDE2KO}$ expressing the same sensor (*Figure 3F*). We next measured FRET changes in MEFs$^{PDE2wt}$ and MEFs$^{PDE2KO}$ expressing H90 or OMM-H90. We found that application of IBMX (100 µM) results in a similar increase in cAMP in the bulk cytosol in the two cell types. In contrast, the cAMP increase at the OMM is significantly higher in MEFs$^{PDE2wt}$ than in MEFs$^{PDE2KO}$ (*Figure 3D,E*). As expected, no response to Bay 60–7550 was detected in either compartment in MEFs$^{PDE2KO}$. Collectively, the above findings indicate that, at least in cardiac myocytes and MEFs, the predominant role of PDE2A is to regulate pmAC-generated cAMP at the OMM, and that any PDE2A activity in the matrix in these cells is below the detection limit of the matrix-H90 sensor. Furthermore, PDE2A2 appears to account for a large fraction (about 85%) of the IBMX-sensitive PDE activity at the OMM.

## Sub-mitochondrial localisation of PDE2A2

To further investigate the submitochondrial localisation of PDE2A2, we performed western blotting analysis of subcellular fractions from NRVM. PDE2A predominantly segregates with the mitochondrial fraction (*Figure 4A*) and treatment with proteinase K (PK, 10 µM), which hydrolyses proteins exposed to the cytosolic side of the OMM, significantly reduced the PDE2A signal, although it did not completely ablate it (*Figure 4A*). As expected, PK treatment completely depleted the mitochondrial fraction of TOM20, a protein exclusively found at the OMM (*Figure 4A*). Similar results were found in HeLa cells overexpressing PDE2A2-GFP (*Figure 4B*). Disruption of the OMM by osmotic shock allowed us to dissect further the localisation of PDE2A at the mitochondria. As shown in *Figure 4C* endogenous PDE2A is clearly detected in the mitoplast fraction obtained from NRVM but is completely depleted on treatment of mitoplasts with PK while with the same treatment the matrix marker COX2 is almost entirely preserved.

To further assess localisation, we performed super-resolution stimulated emission depletion (STED) microscopy of HeLa cells. Cells expressing PDE2A2-GFP were double labelled with anti-GFP antibodies and antibodies to cytochrome C as a marker for the inter membrane space (IMS). As

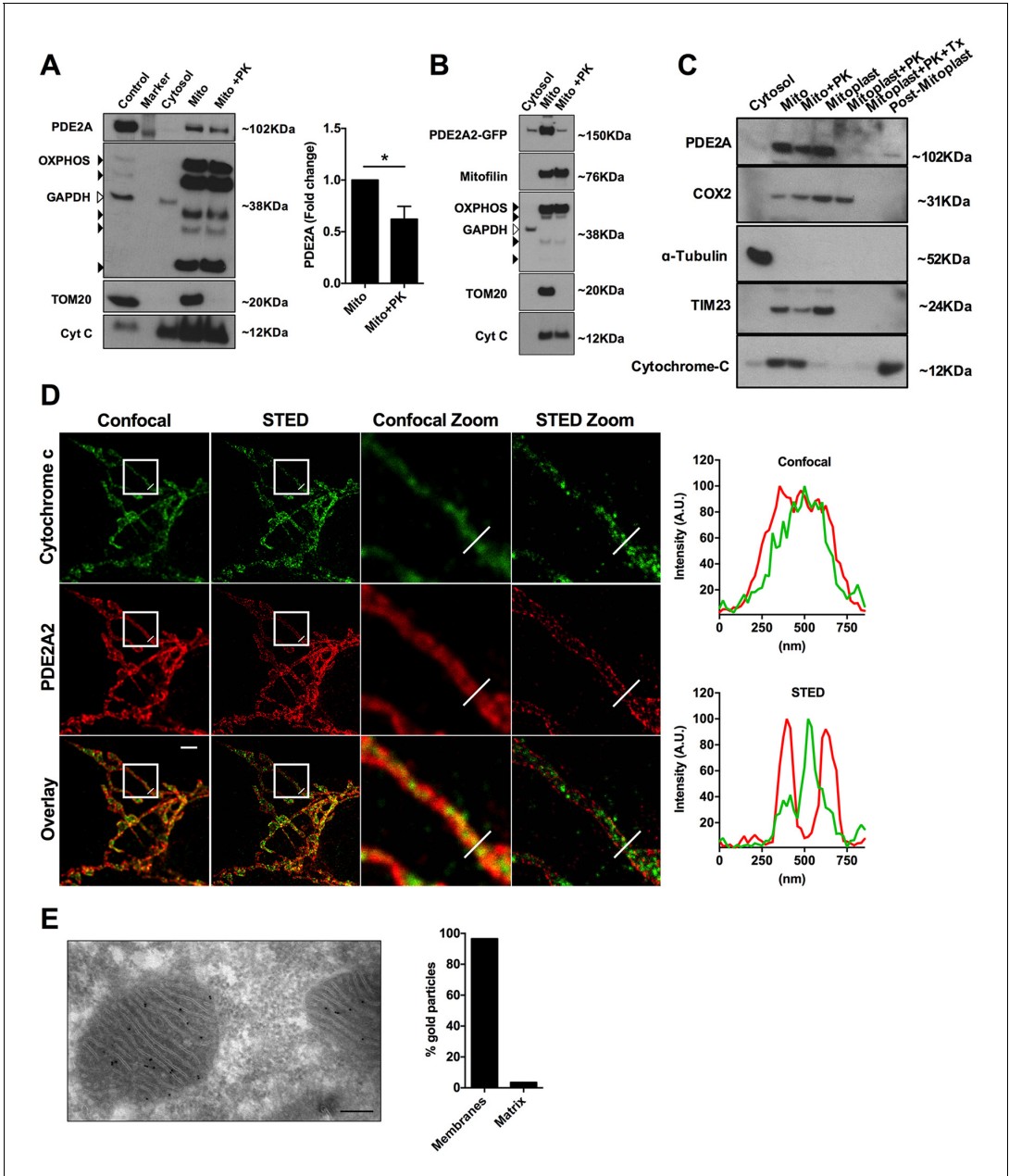

**Figure 4.** Submitochondrial localisation of PDE2A2. (**A**) Representative Western blotting analysis of cytosolic and mitochondrial sub-fractions obtained from NRVM lysates not treated and treated with Proteinase K (10 μM). PDE2A was probed with a PDE2A specific antibody. To assess fraction purity, antibody for OXPHOS subunits (black arrowheads), TOM20 and Cytochrome C were used. GAPDH was used to assess contamination of cytosol in the mitochondrial fraction. The panel on the right shows the quantification from four biological replicates. Student *t*-test was used for statistical analysis. * = p<0.05. (**B**) Representative Western blot of subcellular fractions obtained from Hela cells expressing PDE2A2-GFP. The mitochondrial fraction was either non treated or treated with Proteinase K. PDE2A2 was assessed with a GFP-specific antibody. Mitofilin was probed here in addition to the mitochondrial markers used in **A**). This experiment was repeated twice with similar results. (**C**) Representative Western blot of cytosolic, mitochondrial, mitoplasts and post-mitoplast fractions obtained from NRVM lysates. Samples were either untreated or treated with Proteinase K (PK, 10 μM) or Triton-X plus PK. PDE2A was probed with a PDE2A specific antibody. Cytochrome c oxidase subunit II (COX2) is a marker for mitochondrial matrix; Tubulin is marker for cytosol; TIM23 is marker for IMM; cytochrome-c is a marker for IMS. Blot is representative of three independent experiments. (**D**) Confocal and STED image (first and second column, respectively) of a HeLa cell expressing PDE2A2-GFP and labelled with antibodies to cytochrome c and GFP. Third and fourth columns show magnification of the boxed areas. Plots on the right show average intensity profiles across the indicated mitochondrial tubule section. Cytochrome c profile is in green and PDE2A2 profile is in red. Scale bar: 2 μm. (**E**) Representative electron microscopy image of NRVM probed with PDE2A antibody detected by protein A conjugated with 10 nm gold beads. The count of relative mitochondrial distribution of the immunogold particles is shown on the right (n = 25 mitochondria). Magnification 100X, scale bar: 200 nm.

*Figure 4 continued on next page*

*Figure 4 continued*

The following figure supplement is available for figure 4:

**Figure supplement 1.** Relative localisation of PDE2A2 and either TOM20 or cytochrome c as revealed by STED microscopy analysis.

shown in *Figure 4D*, PDE2A2 appears to be predominantly located at the periphery of mitochondria and to be excluded from the matrix and from the intra-cristae space (where cytochrome c is mostly found). Our analysis shows that the PDE2A2 signal substantially overlaps with that of the OMM marker TOM20 (*Figure 4—figure supplement 1*).

Mitochondrial localisation of PDE2A in NRVM was further assessed by transmission electron microscopy and immunogold labelling. Analysis of immunogold particle localisation shows predominant particle association with mitochondrial membranes (*Figure 4E*), with minimal localisation to the matrix. We found that about 7% of the particles localise to the OMM, resulting in a similar number of particles per unit length of membrane at the OMM and IMM.

Overall, the above data support localisation of PDE2A2 at the OMM and IMM, whereas the enzyme appears to be largely excluded from the intra-cristae space and from the matrix.

## Inhibition of PDE2A protects cardiac myocytes from mitochondria-dependent cell death

Mitochondria elongation has been associated with increased mitochondrial membrane potential ($\triangle\Psi$m) and protection from programmed cell death (*Jahani-Asl et al., 2007*). FCCP application to NRVM loaded with the mitochondrial membrane potential indicator TMRM, revealed a significantly higher $\triangle\Psi$m in myocytes pre-treated with Bay 60–7550 compared to controls (*Figure 5A*). Similar results were found for H9C2 cells (*Figure 5—figure supplement 1*). The higher $\triangle\Psi$m on inhibition of PDE2A was not due to accumulation of protons secondary to reduced ATP production, as mitochondrial respiration was not affected by Bay 60–7550 treatment (*Figure 5B*). To address whether PDE2A2 activity at the mitochondria may modulate the sensitivity of cardiac myocytes to pro-apoptotic stimuli, we exposed MEFs[PDE2KO] and MEFs[wt] to ionomycin (iono), a pro-apoptotic treatment that relies on mitochondrial $Ca^{2+}$ loading and opening of the mitochondrial permeability transition pore. In addition, we used staurosporin (stau), a compound that triggers apoptosis via non-selective kinase inhibition (*Karaman et al., 2008*). We found that, while at the concentration used the two treatments result in comparable death of MEFs[wt], MEFs[PDE2KO] are significantly protected from ionomycin- but not staurosporin-induced cell death (*Figure 5C*). In line with these results, NRVM treated with Bay 60–7550 are protected from ionomycin but not staurosporin treatment, an effect that was not observed when the cells were treated with the PDE3 inhibitor cilostamide (*Figure 5D*). Knock down of PDE2A in NRVM by siRNA[PDE2] resulted in a similar protective effect (*Figure 5E*) and there was no difference between ionomycin and staurosporin treatment when the control siRNA sequence 'siglo' was used (*Figure 5F*). Treatment with BAY60 also completely reversed the effect of ionomycin on $\triangle\Psi$m of NRVM (*Figure 5G*). In addition, TUNEL-staining revealed that the number of apoptotic nuclei induced by ionomycin is significantly reduced in NRVM treated with Bay 60–7550 (*Figure 5H*) compared to controls. Furthermore, while MEFs[PDE2KO] showed significantly less TUNEL-positive nuclei upon treatment with ionomycin compared to staurosporin, MEFs[wt] did not (*Figure 5I*). We further confirmed the requirement of Drp1 for the anti-apoptotic effect of PDE2A inhibition as application of BAY60 was effective at reducing the number of Tunel-positive nuclei resulting from ionomycin treatment in MEFs[wt] but not in MEFs[Drp1KO](*Figure 5J*).

## PDE2A2 regulates mitochondrial morphology and cell survival via modulation of cAMP generated by pmAC

Our data so far support a model where PDE2A2 hydrolyses at the OMM cAMP generated by pmAC. Inhibition of PDE2A2 leads to local activation of PKA, phosphorylation and inhibition of Drp1, mitochondria elongation and cell protection from apoptosis. We used control MEFs (MEFs[WT]) and MEFs derived from sAC knockout mice (MEFs[sACKO]) (*Ramos-Espiritu et al., 2016*) to confirm that the observed effects of PDE2A2 inhibition on mitochondria morphology and function are independent

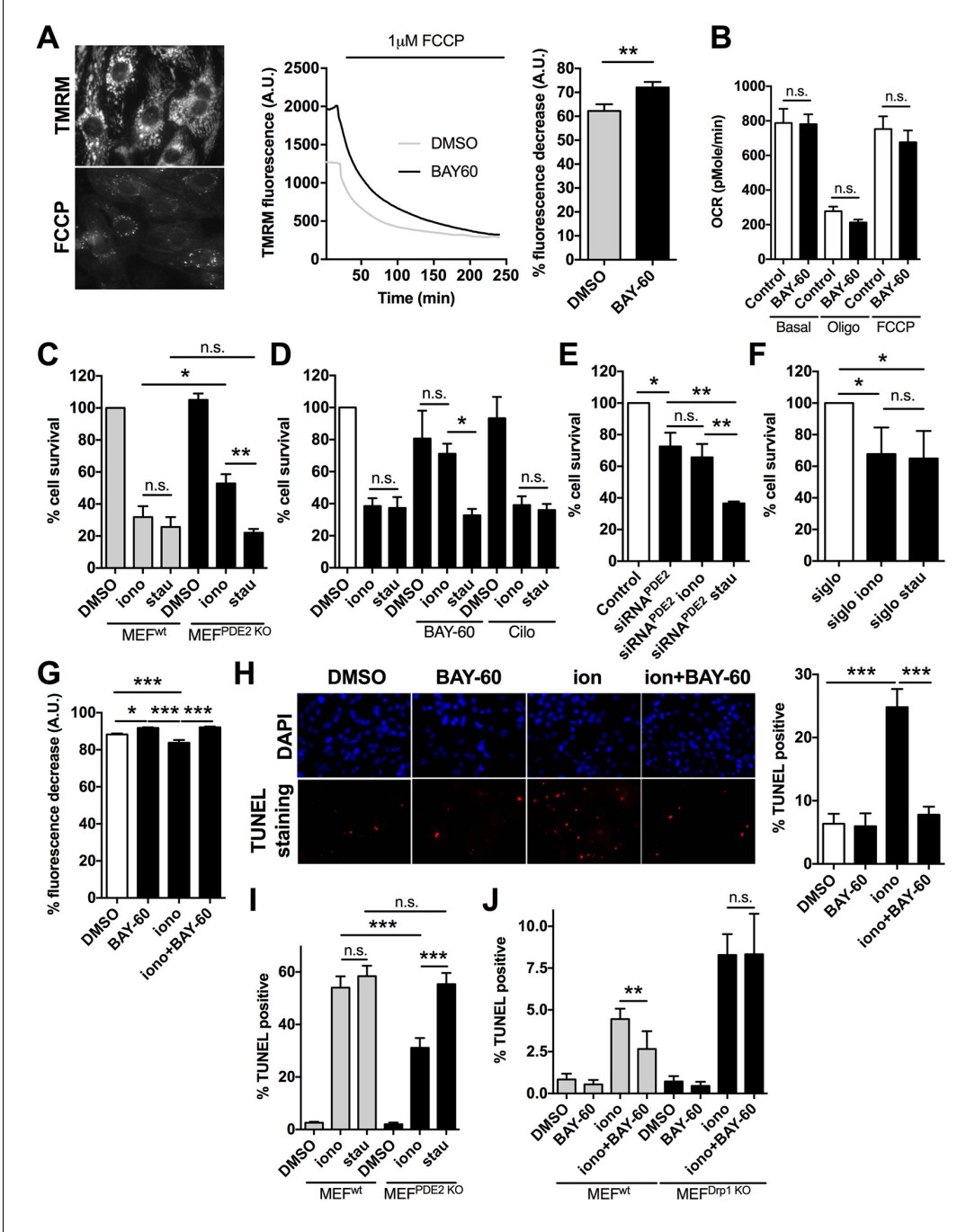

**Figure 5.** PDE2A inhibition affects mitochondrial membrane potential and cell survival. (**A**) Left panel: TMRM fluorescence signal before and after application of 1 μM FCCP to NRVM. Representative kinetics (middle panel) and summary of experiments (right panel, n = 60 cells from three biological replicates) showing the effect of pre-treatment of NRVM with the PDE2A inhibitor Bay 60–7550 (1 μM) compared to control, on the TMRM signal decay after application of FCCP. Values are expressed as percent decrease. (**B**) Oxygen consumption rates (OCR) of NRVM treated with DMSO (control) or with Bay 60–7550 (1 μM) for 24 hr. Basal rates or rates after the addition of 1 μM oligomycin (oligo) or FCCP (100 nM) are the mean ± SEM of three time points in triplicate. Data are from three biological replicates. (**C**) Effect of 10 μM ionomycin or 500 nM staurosporin on survival of MEF[wt] and MEF[PDE2KO] relative to untreated MEF[wt] cells. n = 4 biological replicates, each with eight technical replicates per condition. (**D**) Effect of ionomycin or staurosporin on survival of NRVM treated with ionomycin or staurosporin and in the presence of either the PDE2A inhibitor Bay 60–755 (1 μM) or the PDE3 inhibitor cilostamide (10 μM). Values are expressed relative to untreated controls. n = 6 biological replicates, each with eight technical replicates per

*Figure 5 continued on next page*

*Figure 5 continued*

condition. (**E**) Effect of ionomycin or staurosporin on cell viability of NRVM in which PDE2A was knocked down by siRNA. Values are expressed as relative to untreated and non-transfected cells. n = 4 biological replicates, each with eight technical replicates per condition. (**F**) Effect of ionomycin or staurosporin on cell viability of NRVM transfected with the control siRNA sequence siGLO. n = 4 biological replicates, each with eight technical replicates per condition. (**G**) Summary of experiments (n > 40 cells) showing the effect of acute application of ionomycin (10 µM) on NRVM pre-treated with the PDE2A inhibitor Bay 60–7550 (1 µM) compared to control. Values are expressed as percent decrease of the TMRM signal after application of FCCP. (**H**) Representative images of TUNEL assay performed in NRVM treated with ionomycin (10 µM) or ionomycin (10 µM) plus Bay 60–7550 (1 µM). The bar graph on the right shows the summary data (n = 3 biological replicates, each with two technical replicates per condition). Values are TUNEL positive as percent of DAPI positive nuclei. (**I**) Quantification of TUNEL assay performed in MEF$^{wt}$ and MEF$^{PDE2KO}$ treated with ionomycin (10 µM) or staurosporin (500 nM). n = 3 independent experiments, each with four technical replicates per condition. (**J**) Quantification of TUNEL assay performed in MEF$^{WT}$ and MEF$^{Drp1KO}$ treated with ionomycin (10 µM) or Ionomycin plus Bay 60–7550 (1 µM). n = 3 independent experiments, each with two technical replicates per condition. Student *t*-test was used for statistical analysis in A. ANOVA test with Bonferroni correction was used in all the other cases. *$0.01{\leq}p{\leq}0.05$, **$0.001{\leq}p<0.01$, ***$p<0.001$.

The following figure supplement is available for figure 5:

**Figure supplement 1.** PDE2A affects mitochondrial membrane potential in H9C2 cells.

of any cAMP generated by sAC in the matrix. We found that Bay 60–7550 treatment results in more elongated mitochondria equally in both cell types (*Figure 6A*). In addition, pre-treatment with Bay 60–7550 resulted in a comparable increase in $\triangle\Psi$m in MEFs$^{WT}$ and MEFs$^{sACKO}$ (*Figure 6B*). Similarly, TUNEL analysis showed that both cell types were equally resistant to ionomycin treatment and equally sensitive to staurosporin (*Figure 6C*), confirming that PDE2A2 regulates mitochondria morphology, $\triangle\Psi$m and apoptotic cell death via modulation of cAMP generated by pmAC and not by sAC in the matrix.

## Discussion

The cAMP/PKA signalling cascade controls, within the same cell, multiple functions, affecting processes such as gene transcription, hormone secretion, cell metabolism, cell contraction, cell survival and cell death. This complex and multifaceted functional role is executed with accuracy thanks to a tight temporal and spatial control of protein phosphorylation, which is achieved by compartmentalisation of the molecular components of the signalling cascade. Compartmentalisation of cAMP offers the opportunity to target therapeutically individual cAMP pools, rather than global intracellular cAMP levels, in order to achieve greater efficacy and specificity of intervention (*Maurice et al., 2014*). To assess the translational relevance of such a strategy, a detailed understanding of the organisation and regulation of individual cAMP signalling domains is required. Here we investigated the role of PDE2A in the local regulation of cAMP signalling at the mitochondria and its impact on mitochondrial function.

Numerous studies have linked PDE2A to pathological conditions, including psychiatric, neurological (*Xu et al., 2015*), cardiovascular (*Mehel et al., 2013*), (*Li et al., 2015*) and respiratory (*Zhang et al., 2015*) disorders. Pharmacological inhibition of PDE2A in animal models has been shown to have beneficial effects in a number of conditions, such as cardiac hypertrophy (*Zoccarato et al., 2015*), pulmonary hypertension (*Bubb et al., 2014*) as well as depression, anxiety and cognition deficits in Alzheimer's disease (*Zhang et al., 2015*). A phase I clinical trial examining a PDE2A inhibitor for the treatment of schizophrenia is currently ongoing (*Takeda, 2015*). Despite extensive evidence linking PDE2A to disease and the growing interest of pharmaceutical companies in PDE2A (*Trabanco et al., 2016*), a mechanistic understanding of the role of this enzyme in pathophysiology remains limited.

In this study, we demonstrate that PDE2A2 is part of a novel mitochondrial cAMP/PKA signalling domain. We demonstrate that this domain is fuelled by cAMP generated at the plasma membrane, it drives PKA-dependent phosphorylation of Drp1 and modulates mitochondria dynamics and cell

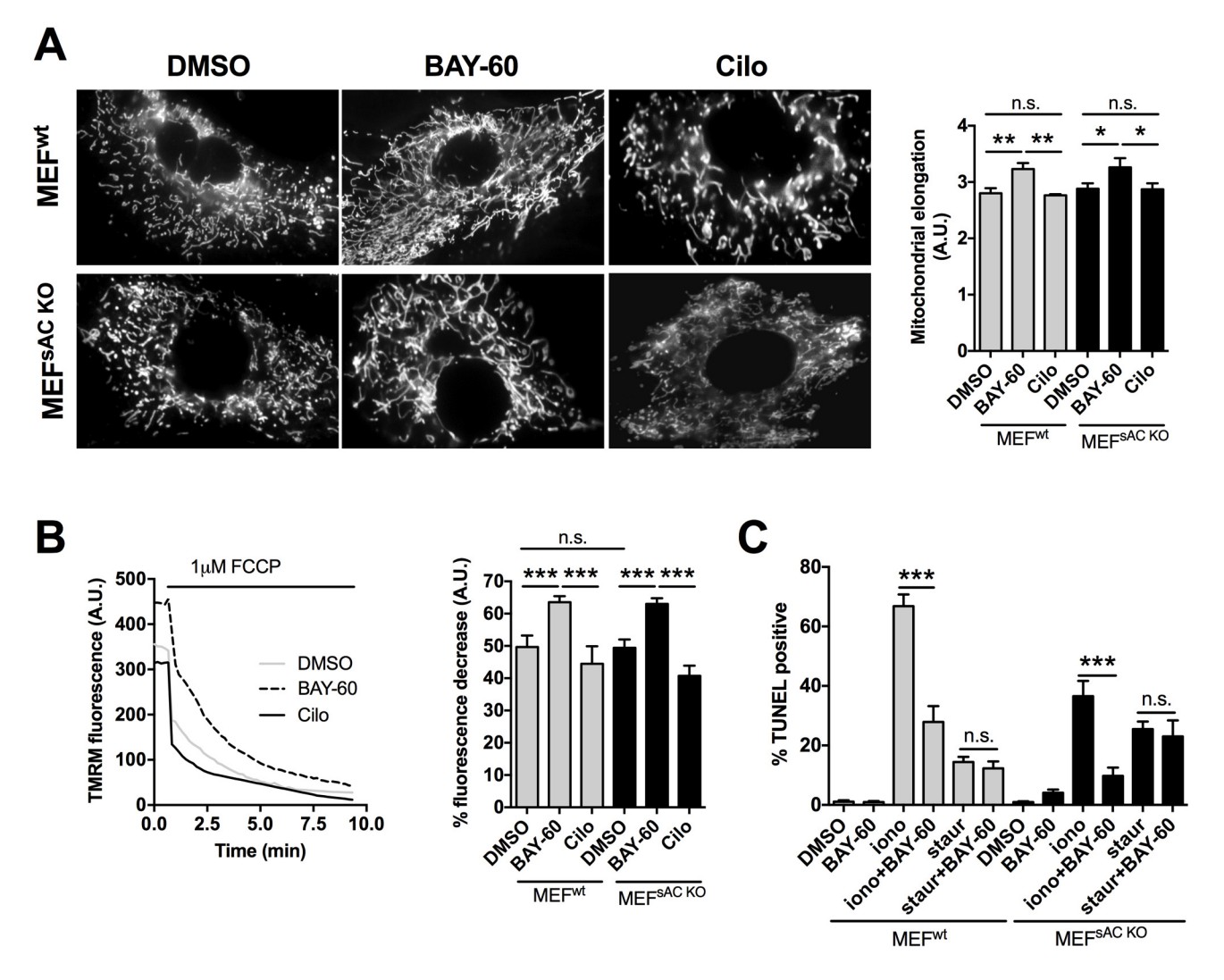

**Figure 6.** The effect of PDE2A2 inhibition on mitochondria morphology, mitochondrial membrane potential and apoptotic cell death is independent of sAC. (**A**) On the left, representative image of MEF[wt] and MEF[sACKO] cells either untreated (DMSO) or treated with Bay 60–7550 (1 µM) or cilostamide (10 µM), and loaded with mitotracker red. The summary analysis of mitochondrial length is shown on the right (n = 22 cells from two independent experiments). (**B**) Analysis of mitochondrial membrane potential measured in MEF[wt] and MEF[sACKO] untreated or treated with Bay 60–7550 or cilostamide and labelled with TMRM. Representative kinetics of TMRM fluorescence decay on application of FCCP to MEF[wt] are shown on the left. Summary data on the right are expressed as percent decrease (n > 50 cells from four independent experiments). (**C**) TUNEL assay performed in MEF[wt] and MEF[sACKO] either untreated or treated with Bay 60–7550, ionomycin, staurosporin, or a combination as indicated. n = 3 independent experiments, each with two technical replicates for condition. ANOVA test with Bonferroni correction was used for statistical analysis. *$0.01 \leq p \leq 0.05$, **$0.001 \leq p < 0.01$, ***$p < 0.001$.

survival. Importantly, we demonstrate that inhibition of PDE2A2 activity results in more elongated mitochondria, increased membrane potential and resistance to pro-apoptotic stimuli. Our STED analysis shows that PDE2A2 localisation at mitochondrial membranes is not homogeneous but shows a dotted appearance (*Figure 4C* and *Figure 4—figure supplement 1B*), which suggests that PDE2A2 may be part of a protein cluster (*Jans et al., 2013*). If confirmed, the identification of PDE2A2 interacting partners at the OMM/IMM may provide an avenue for selective displacement, and therefore specific ablation, of PDE2A activity at this site.

A previous study reported that PDE2A2 localises to the mitochondrial matrix in hepatocytes and neurons where it hydrolyses cAMP generated locally by sAC and regulates ATP synthesis (*Acin-*

*Perez et al., 2011*). In our investigation using primary cells (NRVM and embryonic fibroblasts) and cell lines, we found that the majority of PDE2A2 localises to mitochondrial membranes. Our findings are in line with an earlier report of a cGMP-activated mitochondrial PDE, 20% of which was found to be associated with the OMM and 80% with the IMM (*Cercek and Houslay, 1982*). Although we failed to detect significant PDE2A activity in the mitochondrial matrix using FRET imaging, we cannot exclude that a smaller fraction of the enzyme is present in this compartment. However, we demonstrate that the effect on mitochondria morphology, mitochondria membrane potential and reduced apoptotic cell death observed upon inhibition of PDE2A is not dependent on a sAC/cAMP/PKA/PDE2A signalling domain in the matrix as these effects are unchanged in cells lacking sAC.

Future studies are required to establish the extent to which dysfunction of the cAMP/PKA/PDE2A domain localised at the OMM/IMM underpins PDE2A-related pathological conditions, and whether PDE2A inhibition at this site may be a viable therapeutic avenue. A number of findings warrant these investigations. The positive effect of PDE2A inhibitors on depression/anxiety-like behaviours has been linked to their anti-apoptotic effects (*Ding et al., 2014*). In addition, inhibition of Drp1 was shown to preserve mitochondria morphology and to prevent cell death in a model of myocardial ischemia-reperfusion injury (*Sharp et al., 2014*). Increased PDE2A expression was reported both in rodents (*Hua et al., 2012*) and in human (*Aye et al., 2012*) heart failure and PDE2A inhibition was shown to reduce cardiac hypertrophic remodelling in a mouse model of cardiac hypertrophy (*Zoccarato et al., 2015*), indicating that PDE2A inhibition may be of therapeutic value in this context. Whether the localisation of PDE2A at mitochondrial membranes is related to the anti-hypertrophic effect of PDE2A inhibition in cardiac myocytes remains to be established. However, it is interesting to note that knock down of the mitochondrial fusion protein mitofusin 2, which results in fragmented mitochondria, also leads to cardiac myocyte hypertrophy, suggesting a possible direct link between mitochondria fragmentation and cardiac myocytes hypertrophic growth (*Pennanen et al., 2014*; *Guan et al., 2016*).

Our discovery of a novel cAMP/PKA/PDE2A2 signalling domain located at mitochondrial membranes provides an original mechanistic framework for understanding the involvement of PDE2A2 in pathology and has potential implications for the treatment of mitochondrial diseases.

## Materials and methods

### Reagents, constructs and cell lines

Forskolin, IBMX, Cilostamide, Ionomycin, Pancreatin and Proteinase K were from Sigma-Aldrich (UK). Bay 60–7550 was from Cambridge Bioscience ((UK). Staurosporin was from Calbiochem (CA, USA). Mitotracker red and Mitotracker green were from Life Technologies (CA, USA). SiGLO red transfection indicator and PDE2 3'UTR siRNA were from Dharmacon (CO, USA). Collagenase was from Roche (Switzerland), Laminin (mouse) from BD Biosciences (UK). Transfectin Lipid Reagent was from BioRAD (CA, USA). FuGene6 was from Promega (UK). Phosphate-buffered saline (PBS), DMEM high glucose, MEM199, horse serum, new born calf serum, penicillin/streptomycine, trypsin 0.05% and glutamine were from Invitrogen. Polyplus Jetprime transfection reagent was from VWR (PN, USA). pcDNA3.1 (+) Drp1 V2 (plasmid #44599) and pcDNA3.1 (+) Drp1 V2 S600A (plasmid#46344) were from Addgene (MA, USA). The EGFP-tagged versions of DRP1 and DRP1S600A were generated by subcloning EGFP as BamHI fragment at the N-terminus of DRP1 V2 plasmids. The EGFP-tagged version of PDE2A2 was generated by subcloning PDE2A2 as NheI fragment at the N-terminus of EGFP (pEYFP-C1-EGFP). PDE2A1 (RG235036), PDE2A2 (RG226806) and PDE2A3 (RG207219) in pCMV6-AC-GFP were from Origene (MD, USA). HeLa and H9C2 cells were originally purchased from ATCC, which provides authentication by STR profiling, and tested negative in routinely performed mycoplasma tests in the lab. No cell lines from the list of commonly misidentified cells maintained by the International Cell Line Authentication Committee has been used in this study.

### NRVMs culture and transfection

All procedures were carried out in compliance with the standards for the care and use of animal subjects as stated by the requirements of the UK Home Office (ASPA1986 Amendments Regulations 2012) incorporating the EU directive 2010/63/EU.

Primary cultured NRVM were isolated from 1- to 2 day old Sprague Dawley rats as described (*Zaccolo and Pozzan, 2002*). Ventriculocytes were plated onto laminin coated coverslips or dishes and transfected with Transfectin following the supplier's instructions. Imaging experiments were performed 24–48 hr after transfection.

## MEFs isolation

Pregnant females obtained from crossing of heterozygous PDE2KO mice (line B6;129P2-Pde2a$^{tm1Dgen}$/H, provided by EMMA, UK), were sacrificed after 13 days of gestation (E.13 dpc) and the embryos collected, then placed into separate dishes containing phosphate buffered saline (PBS). A segment of the tail was kept for genotyping, whereas the head and the internal organs were removed. The remaining tissue was washed with PBS and incubated in trypsin at 37°C for 5–10 min. The solution was pipetted until the tissue was completely homogenised and the isolated cells were plated onto T75 flasks. For transfection the cells were plated onto sterile 24 mm$^2$ coverslips and maintained at 37°C for 24 hr. MEFs were transfected with FuGENE 6 or Polyplus Jetprime according to the manufacturer's instructions. Immortalized mouse embryo fibroblasts derived from sAC knockout mice (MEF$^{sACKO}$) (*Ramos-Espiritu et al., 2016*), and the corresponding wild type control were kindly provided by Lonny Levin and Jochen Buck, Weill Medical College, Cornell University, New York. Immortalized mouse embryo fibroblasts lox/lox Drp1 wild type (MEFs$^{wt}$) and Drp1 KO (MEFs$^{Drp1KO}$) were provided by Prof. Naotada Ishihara, Department of Protein Biochemistry, Institute of Life Science, Kurume University, Japan (*Ishihara et al., 2009*).

## Western blotting

NRVM were washed twice with PBS and lysed for 5 min with RIPA buffer containing complete EDTA-free protease inhibitor cocktail tablets (Roche) and PhosSTOP phosphatase inhibitor cocktail (Roche) when required. Pellets were clarified by centrifugation at 10000 rpm for 10 min at 4°C. Protein concentration was determined using the BCA assay (Thermo Scientific, MA, USA) or Bradford assay (Sigma). 30 µg of protein were separated by SDS polyacrylamide gel electrophoresis and subsequently transferred to nitrocellulose membranes (Millipore). Primary antibodies used for detection were: PDE2A (Abcam ab125677), DRP1 (Cell Signalling, 8570), phosphoDRP1 (ser637; Cell Signalling 6319), phosphoDRP1 (ser616; Cell Signalling 3455), GAPDH (Santa Cruz, sc-20357), TOM20 FL-145 (Santa Cruz, sc-11415), Total OXPHOS rodent (Abcam, ab-110413), Mitofilin (Abcam, ab110329)., Tim23 (Santa Cruz, sc-514463), Cytochrome C (Abcam, ab1100325), alpha tubulin (Abcam, ab18251), GFP (Santa Cruz, sc-9996).

## Mitochondria fractionation

Cytosolic and mitochondrial fractions from NRVM were generated using the Qproteome Mitochondria Isolation Kit (Qiagen, 37612), following the manufacturer's instructions. Treatment with 10 µM Proteinase K was performed on the mitochondrial fraction for 25 min at 25°C. Mitochondria were then lysed in RIPA buffer containing protease inhibitor provided by the kit. The same volume (15 µl) of cytosol and mitochondria treated and not treated with Proteinase K were loaded onto gels and blotted for cytosol and mitochondrial markers.

## Mitoplast preparation and protease protection assay

After performing mitochondria fractionation, mitoplasts were prepared as described previously (*Acin-Perez et al., 2011*). Mitochondria were resuspended in 200 µL MS-EGTA (225 mM mannitol, 75 mM sucrose, 5 mM HEPES, 1 mM EGTA, pH 7.4). Water (1/10 vol) and 0.1 mg digitonin were added, and the mixture was incubated on ice for 45 min. Then, KCl (150 mM) was added, followed by incubation for 2 min on ice and centrifugation at 18,000 g for 20 min at 4°C. The pellet containing the mitoplast fraction was re-suspended in 65 µL of 300 mM Tris-HCl, 10 µM CaCl$_2$, pH 7.4. The supernatant containing the post-mitoplasts fraction was precipitated with 12% TCA and centrifuged at 18,000 g for 15 min at 4°C. The pellet was re-suspended in 200 µL acetone and centrifuged at 18,000 g for 15 min at 4°C. Finally, the pellet was re-suspended in 20 µL of 300 mM Tris-HCl, 10 µM CaCl$_2$, pH 7.4

For the protease protection assays, mitoplasts were divided into three tubes: no treatment; proteinase K (PK); PK +1% Triton X-100. Prior to PK treatment, one aliquot of mitoplasts was solubilized

with 1% Triton X-100 for 15 min on ice. 10 μM PK treatment was for 20 min at RT, followed by PK inactivation with protease inhibitor.

## Fluorescence resonance energy transfer (FRET) imaging

FRET imaging experiments were performed 24–48 hr after transfection of cardiomyocytes. Cells were maintained at room temperature in a Ringer modified saline solution (NaCl 125 mmol/L, KCl 5 mmol/L, Na$_3$PO4 1 mmol/L, MgSO$_4$1 mmol/L, Hepes 20 mmol/L, Glucose 5.5 mmol/L, CaCl$_2$1 mmol/L, pH 7.4), and imaged on an inverted microscope (Olympus IX71) using a PlanApoN, 60X, NA 1.42 oil immersion objective, 0.17/FN 26.5 (Olympus, UK). The microscope was equipped with a CoolSNAP HQ (*Archer, 2013*) cooled CCD camera (Photometrics) and a DV (*Archer, 2013*) beam-splitter (MAG Biosystems, Photometrics) for simultaneous imaging of CFP and YFP emissions. The FRET filters used were: CFP excitation filter ET436/20x, dichroic mirror 455DCLP (all from Chroma Technology) in the microscope filter cube; dichroic mirror 505dcxr, YFP emission filter 535/30m, CFP emission filter 480/30m (Chroma Technology) in the beam splitter. Images were acquired using Metafluor software. Changes in cAMP concentration using the H90 (kindly provided by Kees Jalink, Netherlands Cancer Institute), OMMH90 and matrix-H90 (kindly provided by Konstantinos Lefkim-miatis, University of Oxford) sensors were monitored by measuring CFP (480 nm)/YFP (535 nm) fluorescence emission values upon excitation of the transfected cells at 430 nm. FRET changes are expressed as either R/R0, where R is the ratio at time t and R0 is the ratio at time = 0 s, or $\Delta$R/R0, where $\Delta$R = R − R0. Values are expressed as the mean ± SEM.

## Stimulated emission depletion (STED) imaging

Transfected HeLa cells were fixed with 8% paraformaldehyde in TBS (Tris-buffered saline) for 10 min, washed in TBS, permeabilised with 0.5% (v/v) Triton X-100 (TX) in TBS for 5 min and washed in TBS. Non-specific labelling was blocked with a 10 min incubation with 1% normal goat serum (NGS; v/v) in Antibody Diluting Solution (AbDil; TBS 0,1% TX, 2% BSA and 0,1% sodium azide) and all antibodies were diluted in AbDil 1% NGS. Primary antibodies were incubated for 60 min. After 3 × 5 min washes cells were treated with fluorescent-conjugated species-specific immunoglobulins (1:250 in AbDil 1% NGS) for 60 min. After 3 × 5 min TBS washes, coverslips were mounted in Ibidi mounting media and sealed with nail varnish. Primary antibodies used: mouse anti-GFP (Santa Cruz, sc-9996), rabbit anti-GFP (Abcam, ab6556), rabbit anti-TOM20 (Santa Cruz, sc-11415) and mouse anti-cyto-chrome-c (ab110325). Species-specific secondary antibodies STAR RED anti-rabbit and STAR 580 anti-mouse were from Abberior.

Confocal and STED images were obtained using a Leica TCS SP8 3x inverted microscope with a time-gated stimulated emission depletion (STED) module (Leica Microsystems GmbH, Germany), equipped with a white light laser (WLL) and an HC PL APO CS2 100x/1.40 oil objective. Excitation wavelength was 561 and 633 nm and depletion was achieved with a 775 nm laser. The pinhole was set to 0.5 Airy units and the pixel size was 25 nm. Images of 1024 × 1024 pixels were acquired at a scan speed of 600 Hz with the number of line average equal to 16 and frame accumulation of 2. Signal was detected with a hybrid detection system (Leica HyD). For confocal imaging the STED deple-tion beam was turned off and WLL intensity was decreased.

## Electron microscopy

Cryo immuno-EM was carried out as previously reported (*Peters et al., 2001*). Briefly, pellet of cells were fixed with 4% para-formaldehyde and 0.05% glutaraldehyde (both EM grade) for 10 min, and then post-fixed for 30 min with 4% para-formaldehyde alone, at room temperature. After wash with PBS and PBS/glycine the cells were covered with 1% gelatin in PBS 10 min at 37°C and then scraped, pelleted and incubated overnight in sucrose 2.3 M at 4°C and prepared for sectioning (70 nm). The immune labeling was performed on section collected on formvar/carbon coated 100 mesh nickel grids using a polyclonal anti-PDEA2 1:20 (Fabgennix, TX, USA) followed by Protein A conjugated with 10 nm gold beads (Utrecht University, NL). The labeled samples were examined on the electron microscope at 100 kW (Zeiss, Germany).

## Analysis of mitochondria morphology

Quantitative evaluation of mitochondria elongation was performed by deriving a numerical parameter based on the method in (*Spinazzi et al., 2008*). Cells were imaged after incubation with Mitotracker red or green at a concentration of 50 nM for 3 min at 37°C. The mitochondria elongation parameter was computed as $E = F/F_{max}$, where F is MitoTracker fluorescence (background subtracted) averaged within a pixel region of interest (ROI) covering the whole cell area, and $F_{max}$ the highest pixel value in the ROI. Before computing E, 20 iterations of a spatial Gaussian filter were performed to the image to improve E sensitivity. Values indicating mitochondria elongation were expressed as 1/E.

Alternatively (*Figure 1—figure supplement 1*), mitochondrial morphology was assessed according to Song et al, with minor alterations, using OptoMorph software (*Song et al., 2008*). Stacked images underwent maximum projection processing to produce a singular projected image. A threshold was applied to exclude any background fluorescence. The image was then skeletonised so that each mitochondrion could be identified along its longitudinal axis as a one pixel wide object around which an ROI was drawn. Measurements of the length of all the mitochondria within the ROI were then recorded.

## Mitochondrial membrane potential measurements

$3 \times 10^6$ millions NRVM and MEFs were seeded onto six well plates and incubated with DMSO or Bay 60–7550 for 24 hr. Mitochondrial membrane potential ($\triangle \Psi m$) was measured by loading cells with 50 nM tetramethyl rhodamine methyl ester (TMRM, Invitrogen) for 30 min at 37°C. Cells were then washed and imaged with an inverted microscope (Olympus IX71) using a PlanApoN, 60X, NA 1.42 oil immersion objective, 0.17/FN 26.5 (Olympus, UK), equipped with a CoolSNAP HQ (*Archer, 2013*) cooled CCD camera (Photometrics). Basal fluorescence intensity was normalized to the fluorescence intensity collected after acute application of 1 µM FCCP (carbonyl cyanide p-trifluoromethoxyphenhylhydrazone). In RNVM, 10 µM Ionomycin was applied acutely and the cells were imaged to record fluorescence intensity; then FCCP was applied to completely dissipate mitochondrial membrane potential.

## Cell viability assay

NRVM and MEFs were seeded on 96 well plates at $2.5 \times 10^4$ cells/well. Cells were then incubated with Bay 60–7550 1 µM, Cilostamide 10 µM, Ionomycin 10 µM and Staurosporin 500 nM for 24 hr. For PDE2A knockdown with siRNA, cells were transfected with Transfectin and assessed 72 hr after transfection. Cell viability was measured using CellTiter 96 AQueous One Solution Cell Proliferation Assay (Promega), according to the manufacturer's instructions.

## TUNEL assay

$2.4 \times 10^6$ millions NRVM and MEFs were seeded onto 12 well plates and incubated with DMSO, Bay 60–7550, cilostamide, ionomycin or staurosporin for 24 hr. The TUNEL assay was performed using the in situ Cell Death Detection Kit (ROCHE) according to the manufacturer's instructions. The cells were then imaged and the nuclei and fluorescent cells were counted in a blind analysis.

## Oxygen consumption rate measurements

Measurements were performed as previously reported (*Durán et al., 2013*). Briefly, a total of $4 \times 10^4$ cells were plated onto XF24 plates (Seahorse Bioscience, North Billerica, MA, USA) and incubated at 37°C, 5% $CO_2$ for 24 hr. The medium was then replaced with 675 µl of unbuffered Dulbecco's modified Eagle's medium (Seahorse Bioscience) supplemented with 2 mmol glutamine, 25 mmol glucose and 2% fetal bovine serum (pH was adjusted to 7.4 using sodium hydroxide 0.5 M) and cells were incubated at 37°C in a $CO_2$-free incubator. Basal oxygen consumption rate was then recorded using the XF24 plate reader (Seahorse Bioscience). Oxygen consumption rate was normalized to cell number calculated at the end of the experiments. Homogeneous plating of the cells and cell count was obtained by fixing the cells with trichloroacetic acid 10% for 1 hr at 4°C and then staining the fixes cells with 0.47% solution of Sulforhodamine B (Sigma).

## Statistics

Data are presented as means ± standard error of the mean. Graph Pad Prism 5 software was used for all statistical analyses. Student t-test was used to compare two groups of experiments. The comparison of 3 or more groups was performed using ANOVA test with Bonferroni correction. The significance level was set at $p < 0.05$ and the following annotation has been used: $*0.01 \leq p \leq 0.05$, $**0.001 \leq p \leq 0.01$, $***p < 0.001$.

Sample sizes, replicate numbers, and p-values are stated in the figure legends. Biological replicates are independent animals or, for cell lines, independent cultures. Technical replicates are number of cells for FRET experiments, mitochondria morphology and TMRM experiments. For TUNEL experiments and cell viability experiments technical replicates are coverslips from the same culture. Outliers, when encountered, where handled using the Dixon's Q test. For *Figure 3*, cells that did not respond to any treatment, including the saturating stimulus, were excluded.

## Acknowledgements

STED imaging was carried out at the Advanced Light Microscopy Facility at the European Molecular Biology Laboratories, and we are grateful to Dr. Marko Lamp and Dr. Rainer Pepperkok for technical support and assistance. We also thank Prof. Pawel Swietach for access to confocal imaging equipment. This study was supported by the British Heart Foundation (PG/10/75/28537 and RG/12/3/29423) and the BHF Centre of Research Excellence, Oxford (RE/08/004) to MZ and by the Medical Research Council (MR/K501256/1), St. John's College Special Grant and Santander Travel award to MJL.

## Additional information

### Funding

| Funder | Grant reference number | Author |
|---|---|---|
| Medical Research Council | MR/K501256/1 | Manuela Zaccolo |
| British Heart Foundation | PG/10/75/28537 | Manuela Zaccolo |
| BHF Centre of Research Excellence, Oxford | RE/08/004 | Manuela Zaccolo |
| British Heart Foundation | RG/12/3/29423 | Manuela Zaccolo |

The funders had no role in study design, data collection and interpretation, or the decision to submit the work for publication.

### Author contributions

SM, Data curation, Formal analysis, Investigation, Writing—review and editing; MJL, CL, JCC, MW, NCS, RM, MM, Formal analysis, Investigation; GB, Resources, Supervision; NM, Investigation; AS, Supervision; EG, Resources; MB, Software; MZ, Conceptualization, Data curation, Supervision, Funding acquisition, Writing—original draft, Writing—review and editing

### Author ORCIDs

Stefania Monterisi, http://orcid.org/0000-0002-8802-9742
John C Castle, http://orcid.org/0000-0002-6017-7794
Alessandra Stangherlin, http://orcid.org/0000-0001-7296-1183
Mario Bortolozzi, http://orcid.org/0000-0001-7198-9838
Manuela Zaccolo, http://orcid.org/0000-0002-0934-3662

### Ethics

Animal experimentation: All procedures were carried out in compliance with the standards for the care and use of animal subjects as stated by the requirements of the UK Home Office (ASPA1986 Amendments Regulations 2012) incorporating the EU directive 2010/63/EU.

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
