## [Decision Letter]

Thank you for submitting your article "PDE2A2 regulates mitochondria morphology and apoptotic cell death via local modulation of cAMP/PKA signalling" for consideration by *eLife*. Your article has been reviewed by two peer reviewers, and the evaluation has been overseen by a Reviewing Editor and Tony Hunter as the Senior Editor. The reviewers have opted to remain anonymous.

The reviewers have discussed the reviews with one another and the Reviewing Editor has drafted this decision to help you prepare a revised submission..

Summary:

Here, the authors showed that the PDE2A2 cAMP phosphodiesterase is localized to the mitochondria, in part at the outer membrane, and, using either knockdown and re-expression of WT and catalytic mutant PDE2A2 or a selective PDE2A2 inhibitor, they used localized FRET-based cAMP sensor analysis to show that PDE2A2 regulates cAMP levels in the vicinity of mitochondria in cardiac myocytes. Functionally, they showed that inhibition or depletion of PDE2A2 caused mitochondrial elongation, whereas overexpression of PDE2A2 but not PDE2A7 caused mitochondrial fragmentation, concomitant with increased or decreased phosphorylation of S637 in the Drp1 fission protein, a PKA site whose phosphorylation inhibits Drp1 mitochondrial fission activity.

The compartmentalization of cAMP and PKA signaling is highly topical, and the reviewers found your conclusion that PDE2A2 acts at the periphery of mitochondria to control local cAMP concentrations to be of interest. However, they were not fully convinced that your current data provide unequivocal evidence that cAMP levels in the vicinity of mitochondria are regulated by a pool of PDE2A2 on the surface as opposed to by the intramitochondrial PDE2A2 population.

Essential revisions:

1) All the experiments on PDE2A2 sub-mitochondrial localization need to be revisited. The data presented in Figure 4 on the localization of endogenous PDE2A2 are not particularly convincing in terms of an outer membrane localization. Only ~40% of the protein was removed by proteinase K, compared to 100% for TOM20. At face value, these data indicate that the majority of the endogenous protein resides interior to the outer membrane, more consistent with previous studies. Moreover, most of the PDE2A2 in the mitochondrial fraction appears to be immunolocalized on cristae membranes in immunoEM (Figure 4). In addition, although when PDE2A2 was overexpressed as a GFP fusion, it appeared to have a largely outer membrane localization (Figure 4), one could argue that this localization is an overexpression artifact. It is essential to track the remaining ~60% proteinase K-resistant PDE2A2 by biochemical studies to determine its subcellular localization. Other subfractionation approaches are required to inspect mitochondrial localization of PDE2A2 – see for example the swell-contract method used in Murthy and Pande (PNAS 84:378, 1987), and proper quantitation of PDE2A2 protein during fractionation is required. The authors should also discuss more fairly the possibility that a major fraction is inside and only a minor population on the outer membrane. This would not detract from the biological effects they are showing.

2) The authors attribute the elongation of mitochondria in PDE2A2 knockout cells to phosphorylation of Drp1 at Ser637, but unequivocal proof is needed to demonstrate that the protective effect of PDE2A on apoptosis is Drp1 mediated. To this end, they need to repeat the experiments in Figure 6 in cells where Drp1 has been ablated, and WT or non-phosphorylatable Drp1 S637A has been re-expressed, and test if the protection by PDE2A inhibition is additive. At the same time, the levels of phospho-S637 need to be checked by immunoblotting. Ideally, one could show that PDE2A2 knockout cells no longer have elongated mitochondria if a S637E Drp1 phosphomimetic mutant is expressed in the cells.

3) Also in Figure 6, the membrane potential data are puzzling: FCCP is a protonophore and it fully dissipates membrane potential, so the differences upon BAY-60 pretreatment cannot be explained. A correct measurement of membrane potential would require that the authors titrate in oligomycin (to reach maximal TMRM fluorescence) and then FCCP, in conditions where they inhibit the MDR pump for which rhodamines are substrates (for example with verapamil). The difference between steady state TMRM fluorescence and the [max (oligomycin)-min (FCCP)] fluorescence would be an indicator of steady state membrane potential. Yet, a lower membrane potential might just mean more respiring mitochondria, not simply dysfunctional ones. A more meaningful assay would be to correlate cell death protection to preservation of membrane potential.

4) The strongest evidence for a role of PDE2A2 on outer membrane cAMP levels is the data in Figure 3, especially parts D versus E. However, a concern with these experiments is the potential for artifacts from the use of drugs. It is unclear why the nonspecific PDE inhibitor IBMX was used in panel E, instead of BAY60. Also, the behavior of the matrix FRET sensor needs to be compared in PDE2A2 knockout cells.

5) A major concern is how to reconcile the data in this paper with previous work (Acin-Perez et al. Cell Metab. 9:265, 2009) that convincingly showed that PDE is localized to the mitochondrial matrix, where it regulates cAMP levels generated by sAC, which is also localized to the mitochondrial matrix. The production of cAMP activated phosphorylation of respiratory chain subunits and increased oxidative phosphorylation. The authors need to provide a convincing explanation.

[Editors' note: further revisions were requested prior to acceptance, as described below.]

Thank you for resubmitting your work entitled "PDE2A2 regulates mitochondria morphology and apoptotic cell death via local modulation of cAMP/PKA signalling" for further consideration at *eLife*. Your revised article has been favorably evaluated by Tony Hunter (Senior Editor) and three reviewers.

I am pleased to say that in principle your paper is now acceptable for publication in *eLife*. The manuscript has been greatly improved by inclusion of new data to address the original reviewers' concerns, but there are a few minor issues raised by the new reviewer 3 and by the original reviewer 1 that need to be addressed before acceptance, as outlined in the reviews below. Please make the small changes that are requested, and resubmit the paper to *eLife*.

*Reviewer #1:*

The revised manuscript has been improved by the addition of some key new experiments. The results in Figure 4 provide good evidence that the fraction of PDE2A2 within mitochondria are not within the matrix. The new result in Figure 1 is consistent with their model that PDE2A2 inhibition causes mitochondrial elongation through phosphorylation of Drp1 at S637. The results are not particularly striking, however. The figure legend should indicate the meaning of the asterisks, as well as the statistical test used.

*Reviewer #2:*

The revision greatly improved the paper: the conclusions are now fully supported by the experiments presented and this paper would make a very nice addition to the literature.

*Reviewer #3:*

This paper reports the localisation of PDE2A2 at the outer mitochondrial membrane where the authors show that this enzyme contributes to creating a cAMP microdomain that regulates PKA phosphorylation of the mitochondrial fission protein Drp1 (which inhibits its function in fission). Inhibition of PDE2A2 gives mitochondrial elongation, increases membrane potential and protects from apoptosis. The conclusions are supported by pharmacological studies, studies using differentially targeted cAMP FRET reporters and studies in PDE2A2, Drp1 and sAC knockout MEFs. As a new reviewer on the revised version of this manuscript. I have focused on reading the revised paper which I think in general is now a convincing story. I have also gone through the tracked version of the revised manuscript and the reviewers' earlier comments and responses from the authors to these that I in general find have addressed the major comments to the first version.

The remaining comment that I have is that I still (as also brought up in the first review under comment 1 and which has not specifically been commented on by the authors) think the EM immunolocalization still appears to be on cristae membranes rather than OMM, it would be good if the authors could clarify this.

---

## [Author Response]

*Essential revisions:*

*1) All the experiments on PDE2A2 sub-mitochondrial localization need to be revisited. The data presented in Figure 4 on the localization of endogenous PDE2A2 are not particularly convincing in terms of an outer membrane localization. Only ~40% of the protein was removed by proteinase K, compared to 100% for TOM20. At face value, these data indicate that the majority of the endogenous protein resides interior to the outer membrane, more consistent with previous studies. Moreover, most of the PDE2A2 in the mitochondrial fraction appears to be immunolocalized on cristae membranes in immunoEM (Figure 4). In addition, although when PDE2A2 was overexpressed as a GFP fusion, it appeared to have a largely outer membrane localization (Figure 4), one could argue that this localization is an overexpression artifact. It is essential to track the remaining ~60% proteinase K-resistant PDE2A2 by biochemical studies to determine its subcellular localization. Other subfractionation approaches are required to inspect mitochondrial localization of PDE2A2 – see for example the swell-contract method used in Murthy and Pande (PNAS 84:378, 1987), and proper quantitation of PDE2A2 protein during fractionation is required. The authors should also discuss more fairly the possibility that a major fraction is inside and only a minor population on the outer membrane. This would not detract from the biological effects they are showing.*

In our original submission we did clearly state that only a fraction (about 40%) of PDE2A2 resides at the OMM and suggested, based also on our EM analysis and STED imaging, that the remaining 60% of PDE2A2 resided inside the mitochondria at the IMM, as it appeared to be excluded from the intra-cristae space and the matrix. As suggested by the reviewers, to further strengthen this point, we have now performed a more detailed biochemical analysis of submitochondrial fractions. The results are presented in Figure 4 and show that while PDE2A2 can clearly be detected in the mitoplast fraction, it is completely depleted when the mitoplasts are treated with proteinase K (PK), as is TIM23, another protein that localises at the IMM facing the IMS. In contrast, COX2, a protein associated to the IMM but entirely facing the matrix, is not affected by PK treatment, confirming that in our preparation the mitoplasts are intact and that matrix-localised proteins are not affected by PK treatment. Based on these results, and taking into account the imaging data (STED and EM) we conclude that, in NRVM, PDE2A2 localises predominantly at the mitochondria, with a minor fraction being found at the OMM and a larger fraction at the IMM facing the IMS.

*2) The authors attribute the elongation of mitochondria in PDE2A2 knockout cells to phosphorylation of Drp1 at Ser637, but unequivocal proof is needed to demonstrate that the protective effect of PDE2A on apoptosis is Drp1 mediated. To this end, they need to repeat the experiments in Figure 6 in cells where Drp1 has been ablated, and WT or non-phosphorylatable Drp1 S637A has been re-expressed, and test if the protection by PDE2A inhibition is additive. At the same time, the levels of phospho-S637 need to be checked by immunoblotting. Ideally, one could show that PDE2A2 knockout cells no longer have elongated mitochondria if a S637E Drp1 phosphomimetic mutant is expressed in the cells.*

We would like to thank the reviewer for this suggestion as this experiment allows to directly demonstrate the involvement of Drp1 in the effect of PDE2A inhibition. However, we would like to point out that, because inhibition of PDE2A results in an increase of cAMP and PKA-dependent phosphorylation, overexpression of a phospho-null Drp1 is not expected to show additive effect with PDE2A inhibition but is expected to cancel out such effect. Based on this suggestion, to strengthen our conclusion that the effects on mitochondria morphology and cell death that we observe on PDE2 inhibition are dependent on PKA-mediated phosphorylation of Drp1 we performed two completely new sets of experiments using MEF^wt^ and MEF^Drp1KO^. In Figure 1 we show that the effect of PDE2A inhibition on mitochondria elongation is not detectable in MEF^Drp1KO^ cells. The ability of PDE2A inhibition to induce elongation of mitochondria is restored in MEF^Drp1KO^ that express wt Drp1 but not a Drp1 mutant that cannot be phosphorylated by PKA. In addition, in Figure 5 we show that while we confirm in wt MEFs that BAY60 protects the cells from ionomycin-induced apoptosis, this effect is lost in MEF^Drp1KO^, confirming that for PDE2A inhibition to exert a protective effect, Drp1 is required. Expression of a Drp1 phosphomimetic in PDE2AKO cells is not expected to reverse the phenotype (elongated mitochondria) of PDE2AKO cells as lack of PDE2A leads to increased phosphorylation of Drp1 and phosphomimetic Drp1 (always inactive) would also result in elongated mitochondria. We therefore did not perform this experiment.

Unfortunately, we were not able to confirm by western blot the level of phosphorylated Drp1 in the experiments with expression of Drp1 wt in MEF^Drp1KO^. We believe this depends on the low fraction of Drp1 that is phosphorylated in MEFs. In support of this interpretation is the fact that we could only detect a very weak signal for phospho-Drp1 when the recombinant Drp1 was pulled-down from lysates of MEFs treated with saturating concentrations of Forskolin and IBMX (See Figure 7). Cells treated with Bay60 did not show any signal at all.

Author response image 1.**DOI:**
http://dx.doi.org/10.7554/eLife.21374.014

*3) Also in Figure 6, the membrane potential data are puzzling: FCCP is a protonophore and it fully dissipates membrane potential, so the differences upon BAY-60 pretreatment cannot be explained. A correct measurement of membrane potential would require that the authors titrate in oligomycin (to reach maximal TMRM fluorescence) and then FCCP, in conditions where they inhibit the MDR pump for which rhodamines are substrates (for example with verapamil). The difference between steady state TMRM fluorescence and the [max (oligomycin)-min (FCCP)] fluorescence would be an indicator of steady state membrane potential. Yet, a lower membrane potential might just mean more respiring mitochondria, not simply dysfunctional ones. A more meaningful assay would be to correlate cell death protection to preservation of membrane potential.*

The presentation of our data on mitochondria membrane potential in the first submission was unclear and has created a misunderstanding. The representative traces shown in our original Figure 5 and Figure 6 were normalised to the basal fluorescence values, which is what we did to calculate the percent drop in membrane potential on FCCP application. We realise now that this way of presenting the data is misleading as it looks like FCCP is not dissipating completely the membrane potential in all treatment groups. We apologise for the confusion that this has generated. We have now corrected this and the new Figure 5 and Figure 6 have raw, non-normalised data, which clearly show that treatment with PDE2A2 inhibitor results in higher membrane potential compared to untreated cells and that FCCP, as expected, dissipates the membrane potential completely in all conditions.

We thought however that the concern raised by the reviewers about the possibility that a lower membrane potential could result from mitochondria that respire more and, vice versa, that a higher membrane potential may result from mitochondria that respire less, was an important point to address. We therefore performed Seahorse measurement to assess mitochondria respiration in cardiac myocytes treated with the PDE2A inhibitor and found no significant difference compared to untreated myocytes (data are presented in Figure 5), confirming that the increased membrane potential on Bay 60-7550 application is not due to deficient respiration. Finally, to try and correlate cell death protection to preservation of membrane potential, we assessed whether PDE2A inhibition would preserve membrane potential in cardiac myocytes subjected to pro-apoptotic stimuli. As shown in the new Figure 5, while ionomycin treatment results in a significant reduction in the mitochondrial membrane potential, Bay 60-7550 completely restores the membrane potential to control values.

*4) The strongest evidence for a role of PDE2A2 on outer membrane cAMP levels is the data in Figure 3, especially parts D versus E. However, a concern with these experiments is the potential for artifacts from the use of drugs. It is unclear why the nonspecific PDE inhibitor IBMX was used in panel E, instead of BAY60. Also, the behavior of the matrix FRET sensor needs to be compared in PDE2A2 knockout cells.*

We have now included experiments where we show the effect of Bay 60-7550 on cAMP levels both in the cytosol and OMM of both MEF^wt^ and MEF^PDE2KO^ (new Figure 3). The data show that, as expected, inhibition of PDE2A does not generate an appreciable response in both cytosol and mitochondria of MEF^PDE2KO^ cells. In addition, while IBMX treatment results in the same cAMP increase in the cytosol of MEF^wt^ and MEF^PDE2KO^, the effect of IBMX is almost completely abolished at the OMM in MEF^PDE2KO^ cells compared to MEF^wt^, further supporting that PDE2A is the predominant PDE at the mitochondria. In addition, we now provide data showing the response of matrix-H90 expressed in both MEF^wt^ and MEF^PDE2KO^. These data are presented in Figure 3 and show that the FRET change on PDE2A inhibition is indistinguishable in the two cell types, confirming that the activity of any matrix-localised PDE2A is not detectable with this sensor.

*5) A major concern is how to reconcile the data in this paper with previous work (Acin-Perez et al. Cell Metab. 9:265, 2009) that convincingly showed that PDE is localized to the mitochondrial matrix, where it regulates cAMP levels generated by sAC, which is also localized to the mitochondrial matrix. The production of cAMP activated phosphorylation of respiratory chain subunits and increased oxidative phosphorylation. The authors need to provide a convincing explanation.*

In this study we present multiple lines of evidence that consistently support localisation of PDE2A2 at mitochondrial membranes (both OMM facing the cytosol and IMM facing the IMS). Biochemical fractionation, super resolution microscopy and electron microscopy all confirm predominant localisation of PDE2A2 at these sites. Our FRET imaging experiments also exclude that a major fraction of PDE2A2 localises at the matrix as the response detected at this site on inhibition of PDEs (both with the non-selective inhibitor IBMX and with the PDE2-selective inhibitor BAY60) using a matrix-targeted reporter is the same in MEFs wt and MEF^PDE2KO^. By contrast, both IBMX and BAY60 generate a robust response at the OMM in MEFs wt but no response at this site in MEF^PDE2KO^. Although it is possible that a fraction of PDE2A2 is localised in the matrix in cardiac myocytes and fibroblasts we were not able to detect it with any of the approaches we used.

We would also like to point out that the data presented in Acin-Perez et al., Cell Metab 2009 are not incompatible with localisation of PDE2A at mitochondrial membranes. In that study the evidence that PDE activity is in the matrix is based on an experiment where exogenous cAMP is added to a permeabilised mitoplast preparation with or without addition of the non-selective PDE inhibitor IBMX (Acin-Perez et al. Cell Metab. 9:265, 2009, Figure 2). The first consideration is that the study does not address specifically the role of PDE2A. Therefore, it is possible that another PDE, different from PDE2A2, may localise to the matrix and be responsible for hydrolysis of cAMP. We also observe that, based on the data presented, it is not possible to exclude that the cAMP hydrolysing activity is due to a PDE localised to the IMM facing the IMS (a PDE localised at that site would be present in their preparation). To firmly conclude that the observed mitoplasts cAMP hydrolytic activity is due to a PDE localised in the matrix one would need to assess the effect of IBMX in intact mitoplast and in mitoplasts that have been treated with PK prior to sonication.

In a subsequent study (Acin-Perez et al. J Biol Chem 286, 2011) the same group addresses directly the submitochondrial localisation of PDE2A2 using a biochemical approach. The key data, (Acin-Perez et al. J Biol Chem 286, 2011, Figure 2) in our opinion do not incontrovertibly demonstrate that PDE2A2 is exclusively localised to the mitochondrial matrix for the following reasons: 1) there is a reduction in the PDE2A signal in the intact mitochondrial fraction after treatment with PK. This suggest that a fraction of PDE2A2 is indeed at the OMM; ii) although the PDE2A2 signal persists in the mitoplast preparation after treatment with PK, its intensity is reduced compared to untreated mitoplasts. Notably, the signal for TIM23, a protein that is localised to the IMM facing the IMS, also persists in PK-treated mitoplasts, suggesting that the enzymatic treatment is possibly not completely effective. This, rather than predominant localisation in the matrix, could account for the persistent PDE2A2 signal in the PK-treated mitoplast fraction. Therefore, in our opinion the data presented in that study do not exclude a localisation of PDE2A2 at the IMM facing the IMS.

Another consideration is that the predominant localisation of PDE2A2 may be different in different cell types. Acin-Perez et al. investigate the role of sAC in the regulation of oxidative phosphorylation in liver and brain cells, whereas our study is focused on cardiac myocytes and embryonic fibroblasts. It is interesting to note that inhibition of PDE2A results in increased respiration in brain mitochondria (Acin-Perez et al. J Biol Chem 286, 2011, Figure 4) whereas our data in primary cardiac myocytes show no effect of BAY60 on oxygen consumption (our study, Figure 5). It is therefore possible that the distribution of PDE2A2 in different submitochondrial compartments varies in different cell types, possibly resulting in different functional effects.

[Editors' note: further revisions were requested prior to acceptance, as described below.]

*Reviewer #1: […] The new result in Figure 1 is consistent with their model that PDE2A2 inhibition causes mitochondrial elongation through phosphorylation of Drp1 at S637. The results are not particularly striking, however. The figure legend should indicate the meaning of the asterisks, as well as the statistical test used.*

We now include the following information in the legend to Figure 1: Anova test with Bonferroni correction was used to compare groups of three. *0.01≤p≤0.05, **0.001≤p<0.01, ***p<0.001.

*Reviewer #3:*

*[…] The remaining comment that I have is that I still (as also brought up in the first review under comment 1 and which has not specifically been commented on by the authors) think the EM immunolocalization still appears to be on cristae membranes rather than OMM, it would be good if the authors could clarify this.*

In mitochondria the surface of the IMM is several times greater than the surface of the OMM. This is reflected in the mitochondria sections captured by EM images, where the length of the IMM largely exceeds that of the OMM. It is therefore to be expected that the number of particles detected at the OMM is significantly smaller than the number found at the IMM. We did perform a more detailed analysis of the number of particles at the OMM and found that this is about 7% of the total. A rough estimation of the particle density per unit of membrane indicates that there is no difference between OMM and IMM. To clarify this point we have now added a sentence to the subsection “Sub-mitochondrial localisation of PDE2A”, including this information.